# From Tokens to Meaning: LLMs and LVLMs Require Semantic-Level Uncertainty

## Abstract

This position paper argues LLM and LVLM reliability should go beyond hallucinations and integrate uncertainties. Furthermore, the commonly used token-level uncertainty is insufficient and semantic-level uncertainty is key. Token-based criteria, such as next-token entropy or maximum probability, work well in closed-world tasks where the output space is predefined and bounded. However, foundation models increasingly operate in open-world settings. The space of answers is unbounded and queries may involve unseen entities, ambiguous phrasing, or complex reasoning. In such cases, token-level confidences may be misleading; outputs with high probability may be semantically wrong, irrelevant, or hallucinatory.

We advocate shifting toward **semantic-level uncertainty** to capture uncertainty in the meaning of generated outputs. By doing so, we can better characterize phenomena such as ambiguity, reasoning failures, and hallucination. We further argue that semantic uncertainty should become the primary lens through which we assess the reliability of foundation models in high-stakes applications, enabling more faithful, trustworthy, and transparent AI systems.

## 1 Introduction

Large Language Models (LLMs) (Brown et al., 2020; Achiam et al., 2023; Touvron et al., 2023; Jiang et al., 2024) and Large Vision-Language Models (LVLMs) (Zhou et al., 2024; Bommasani et al., 2021) have remarkable generalization capabilities. They have quickly shifted from research prototypes to products used by millions of people daily. They are used in a wide range of high-stake settings, such as as medical diagnosis, autonomous driving, or legal decision support and personal health recommendations, education, etc. This raises profound questions about their reliability and safety as they exhibit provide confident answers without an explicit notion of how uncertain the answers may be. This drawback is often generically denoted as hallucinations can emerge in diverse and unpredictable ways that are difficult to anticipate or control fully.

Classic machine learning responds to this issue with uncertainty estimation and robust prediction (Kendall & Gal, 2017; Senge et al., 2014; Hüllermeier & Waegeman, 2021). It distinguishes two main categories of uncertainty: (i) **aleatoric uncertainty**, which arises from inherent randomness or noise in the data; it is irreducible even with infinite data, as it arises from e.g., ambiguity in labeling or stochastic real-world effects; (ii) **epistemic uncertainty**, due to lack of knowledge or limited model capacity; such uncertainty is reducible with more data or better modeling. When extending these ideas to LLMs and LVLMs, important complications arise. These models must handle multi-modal inputs (e.g., images and text) and generate free-form text outputs. Since many different *token* sequences can express the same meaning, uncertainty cannot be fully understood at the *token level* alone, making classical definitions harder to apply directly.

Uncertainty for LLMs and LVLMs is often quantified at a *token level* based on next-token probabilities or entropies, since the models conveniently provide output token likelihoods. Consider the query *"What is the capital of Switzerland?"*, to which the model responds *"I am not sure but I think it is Zurich."* At the token level, the model may exhibit low entropy, indicating that it is *confident* in producing this particular sequence of words. Yet from a semantic standpoint, the model has expressed uncertainty about the question. This reveals a gap: token-level uncertainty does not necessarily align with the uncertainty expressed by the model (or perceived by the user).

In the past years, the reliability of LLMs and LVLMs has been studied largely through the lens of hallucinations. LLMs and LVLMs often produce outputs that are fluent and plausible but factually incorrect or inconsistent with the input. For example, given a medical image, an LVLM might confidently generate *"Two tumors are visible"* when asked *"How many tumors are present in this scan?"*, even if one or no tumor is present. Token-level measures may misleadingly suggest high certainty because the sequence is produced with high probability. This issue becomes clearer when contrasting *closed-world* and *open-world* learning scenarios. In a closed-world setting, the model is trained and evaluated on a well-defined set of classes or tasks, and all possible outputs are assumed to lie within this known universe. For instance, in regular image classification on ImageNet (Russakovsky et al., 2015), every test image is assumed to belong to one of the fixed 1000 categories. In such settings, token-level uncertainty (e.g., the entropy of predicted class labels or next-token probabilities) often provides a reasonable proxy for model confidence (Hu et al., 2023), since the space of outcomes is bounded and well-specified. In contrast, LLMs and LVLMs can operate in inherently open-world environments. Users may ask about new concepts, unseen entities, or ambiguous queries, and the space of possible answers is effectively unbounded.

To address this issue, semantic entropy (Farquhar et al., 2024; Kossen et al., 2024) can measure semantic uncertainty at a sequence level by leveraging an external model to predict the entailment of multiple generated answers and quantify uncertainty from the clusters of answers. Semantic uncertainty presents a significant step forward but is still not widely adopted.

> **Position**
>
> Our position is that traditional token-level uncertainty is not sufficient for LLMs and LVLMs. We argue that the field must shift its focus toward *semantic-level uncertainty*, which reflects uncertainty in the meaning supported by the sequence of output tokens. Moreover, reliability of LLMs and LVLMs should go beyond the generic traditional sources of uncertainty (aleatoric, epistemic) and beyond the widely studied but inconsistently defined hallucination problem. Reliability of such models should take into consideration the specific sources of uncertainty that arise for multi-modal inputs and textual outputs and tasks. Semantic-level uncertainty can provide valuable insights to better capture phenomena such as ambiguity, hallucination, and epistemic limits that token-level cannot.

**Our Contributions.** This paper makes three contributions:

1. We propose a taxonomy for uncertainty quantification in LLMs and LVLMs and explicitly link it to hallucination.
2. We introduce a formalism to analyze how uncertainty evolves with the context provided to the model.
3. We discuss the limitations of semantic uncertainty and outline directions for future work.

## 2 TOKEN-LEVEL VS. SEMANTIC-LEVEL UNCERTAINTY

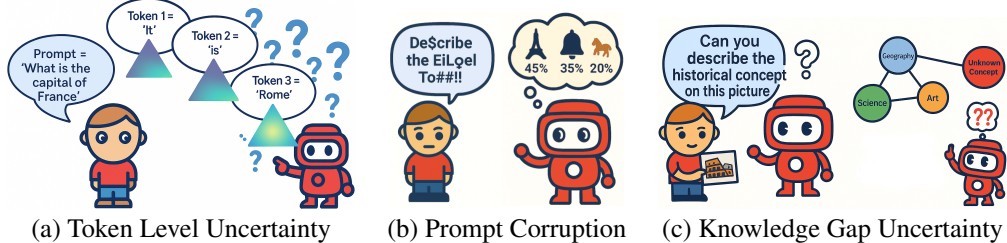

(a) Token Level Uncertainty     (b) Prompt Corruption     (c) Knowledge Gap Uncertainty

Figure 1: **Examples of uncertainty sources/types in LLMs and LVLMs.** (a) Token-level uncertainty: the model is unsure about the next token. (b) Prompt corruption: errors in the prompt make the answer unreliable. (c) Knowledge gap: the model is uncertain when the prompt asks for information it does not know.

## 2.1 Uncertainty from Classic Learning Models to LVLMs and LLMs

Let us begin with a simple example from classic machine learning. Assume we want to learn a model $f_\omega$ that predicts tomorrow's temperature from past weather data $X$ (humidity, wind, previous temperature). The target is $Y$, the actual temperature. The goal of model $f$ is to approximate the relationship between $X$ and $Y$.

For example, given past features $\boldsymbol{x}$, the model may predict $y = 35°$C in Tokyo. But how certain is this prediction? Could it just as well be 33 or 37? What is the *uncertainty* associated with the prediction (Philip Dawid & Vovk, 1999)? Point estimates alone do not answer this.

A probabilistic model instead predicts a distribution $p(Y|\omega, \boldsymbol{x})$, assigning high probability to 35 but also some mass to nearby values. This distribution provides uncertainty estimates, which are crucial for decision-making in weather-sensitive tasks.

With large vision-language models (LVLMs), the problem is even more complex. An LVLM $f_{\boldsymbol{\omega}}$ takes both an image $\boldsymbol{x}^{\text{img}}$ (e.g., a satellite photo) and a text prompt $\boldsymbol{x}^{\text{txt}}$, and outputs text $\boldsymbol{y}^{\text{txt}}$:

$$\boldsymbol{y}^{\text{txt}} = f_{\boldsymbol{\omega}}(\boldsymbol{x}^{\text{img}}, \boldsymbol{x}^{\text{txt}}).$$

For example, prompting

$$\boldsymbol{x}^{\text{txt}} = \text{"What is the temperature of Tokyo from this satellite image?"}$$

might yield

$$\boldsymbol{y}^{\text{txt}} = \text{"35"}.$$

Here, uncertainty arises from many sources: Did the model interpret "temp" correctly? Does it know which part of the image is Tokyo? Such factors make uncertainty estimation especially important in LVLM predictions.

## 2.2 Sources of Uncertainty in LLMs and LVLMs

Like classic deep neural networks, both LLMs and LVLMs inherit well-known sources of uncertainty from data and from the model. However, their multimodal nature, open-world deployment, and text-based inputs introduce extra sources of uncertainty unique to these models (Xia et al., 2025; Liu et al., 2025; Karim et al., 2025). We list and illustrate the most important sources below.

**1. Prompt Corruption (Zhu et al., 2023).** LLMs and LVLMs are sensitive to small, semantically meaningless variations in the prompt (e.g., reordering words, synonyms, or rephrasing). Despite conveying the same intent, such perturbations can drastically alter the model's behavior. This form of perturbation has been studied under prompt robustness and adversarial prompting (Zhao et al., 2021; Abbasi Yadkori et al., 2024), indicating brittleness at the boundary between syntactic and semantic comprehension.

**2. Knowledge gaps and training coverage (Ahdritz et al., 2024).** LLMs have a fixed knowledge cutoff date, and both LLMs and LVLMs may encounter entities, events, or concepts not represented in their training data. For instance, asking an LLM trained before 2023 about the winner of the 2025 World Cup forces it to extrapolate. Similarly, an LVLM trained on natural images may perform poorly on medical X-rays or satellite images, leading to epistemic uncertainty.

**3. Prompt Underspecification (Yang et al., 2025).** Unlike classic models, LLMs and LVLMs rely on natural language prompts, which are often incomplete or vague. An underspecified prompt does not provide enough information to uniquely determine what the model should produce. For instance, the query *"Who is the president?"* lacks essential context: the president of which country or organization, and at what year? Such incompleteness forces the model to guess the user's intent, thereby introducing uncertainty. Prompt underspecification thus represents a large source of unreliability in generative models.

**4. Reasoning complexity and compositionality**(Chen et al., 2023). Multi-step reasoning tasks amplify uncertainty. For example, answering *"If John is older than Mary, and Mary is older than Paul, who is the youngest?"* requires chaining logical steps. Errors in intermediate steps accumulate and lead to uncertain or incorrect outputs. For LVLMs, questions such as *"Is the same person in both images?"* require both visual recognition and logical comparison.

**5. Multimodal grounding errors (LVLM-specific)**(Lu et al., 2023). LVLMs need to correctly match words with the right parts of an image. Uncertainty arises when this link, called grounding, fails. This can happen if the model relies too much on text patterns and ignores the image (Schrodi et al., 2024; Kaduri et al., 2025), or if it struggles to locate objects correctly (e.g., saying "the cat on the left" when it is actually on the right). Grounding errors may also occur when the model misunderstands relationships in the text, such as "the cat sitting on the grass." These cases show uncertainty in the model's ability to connect vision and language.

**6. Decoding randomness**(Abbasi Yadkori et al., 2024). LLMs and LVLMs often rely on stochastic decoding strategies such as sampling or nucleus sampling. Different runs may produce different outputs, leading to variability in model behavior. Although some of this variability is unimportant (e.g., different phrasings of the same meaning), in other cases it reflects true model uncertainty.

In summary, LLMs and LVLMs inherit classic sources of uncertainty such as data uncertainty and model uncertainty, but also exhibit new ones related to prompt-driven interaction, multimodal grounding, and large-scale open-world usage. Quantifying and disentangling these sources is essential for building trustworthy AI systems.

## 2.3 DEFINITION OF THE TYPES OF UNCERTAINTY

To better capture these subtleties, we distinguish between **token uncertainty** and **semantic uncertainty** similarly to Kuhn et al. (2023):

**Definition 2.1** (Token-level Uncertainty). Token-level uncertainty refers to the uncertainty associated with individual output units—such as words in language models or image patches in vision models. This type of uncertainty closely aligns with classic notions of uncertainty in deep learning, where predictions are made at a fine-grained level and modeled through probabilistic outputs.

**Definition 2.2** (Semantic-level Uncertainty). Semantic-level uncertainty refers to the uncertainty over the *meaningful interpretation* of a generated or expected output, considering the alignment with underlying user intent, world knowledge, and visual understanding.

Token uncertainty is inherently local and syntactic—it quantifies variability at the level of surface form. While useful in evaluating next-token prediction models, it fails to capture the broader picture when dealing with generative systems where many different but equally valid responses are possible. For instance, the prompt *"Describe the emotion of the person in the image"* may elicit several plausible completions depending on context and interpretation, all of which may be semantically correct but involve distinct token sequences. Relying solely on token entropy can thus overestimate uncertainty where true semantic ambiguity is low.

Token-level uncertainty—while useful for detecting sampling noise or entropy spikes—does not distinguish between multiple correct semantic outcomes and genuine confusion. A model may exhibit high token entropy even when it has full semantic clarity (e.g., choosing between "happy," "joyful," and "cheerful"). Conversely, it may output a low-entropy response that is semantically wrong due to overconfidence.

Hence, we argue that semantic-level uncertainty—centered on the *meaningfulness and correctness of the output in context*—is better suited for evaluating the reliability of LVLMs and LLMs.

## 2.4 HALLUCINATION AND UNCERTAINTY

Hallucinations in LLMs and LVLMs have been extensively studied (Huang et al., 2025; Filippova, 2020; Ji et al., 2023; Liu et al., 2024; Zhang et al., 2024), since they represent one of the most fundamental and persistent problems of these models. The term *hallucination* is evocative: it suggests

that the model "perceives" or produces content that is not grounded in reality, much like a human hallucination reflects experiences disconnected from the external world.

In the context of open-world models such as LLMs and LVLMs, hallucinations are not rare anomalies but rather a core difficulty. Because these models are often deployed in open-world settings where the range of possible outputs is extremely broad, hallucinations can arise naturally from the mismatch between model knowledge, reasoning capacity, and user expectations.

A number of taxonomies have been proposed in the literature to differentiate types of hallucinations in generative models. In this work, we adopt the classification of Huang et al. (2025), who categorize hallucinations into two primary types: *factuality hallucinations* and *faithfulness hallucinations*.

**Factuality hallucinations.** Factuality hallucinations occur when the generated content diverges from verifiable external knowledge about the world. In other words, the model produces a statement that is fluent and plausible but factually incorrect. For example, when asked *"Who is the Chancellor of Germany in 2025?"*, an LLM might confidently reply *"Olaf Scholz"*, even though the mandate of Olaf Scholz has ended. Such hallucinations are dangerous because they present incorrect information as if it were true, and users may not have the means to easily detect the discrepancy.

**Faithfulness hallucinations.** Faithfulness hallucinations, in contrast, emphasize a divergence between the model's output and the specific input prompt or context. Here, the generated response may be internally coherent but fails to remain faithful to the provided input, instructions, or reasoning trajectory. For example, if an LVLM is shown a chest X-ray and asked *"Is there fluid accumulation?"*, but instead replies *"There is a tumor in the left lung"*, the output is not only medically incorrect but also unfaithful to the user's original query. These errors are less about real-world factual correctness and more about the model's ability to align its generation with the conditioning input and its own internal reasoning trajectory.

**Linking hallucination and uncertainty.** The sources of hallucination are deeply linked with the sources of uncertainty discussed in Subsection 2.2. We can group them into three broad categories:

1. **Training data.** Datasets inevitably encode biases, coverage limitations, and temporal inconsistencies. For example, societal biases present in large text corpora can lead to stereotypical hallucinations (e.g., associating certain professions only with one gender). Long-tail or niche knowledge may be underrepresented, leading the model to "fill in the gaps" with fabricated details. Similarly, outdated data causes the model to hallucinate facts that were true at training time but are no longer accurate.

2. **Training procedure.** Hallucinations may also stem from how the model is trained. Pre-training on large-scale corpora with noisy or unreliable text introduces factuality errors. Fine-tuning or reinforcement learning from human feedback (RLHF) (Ouyang et al., 2022) can exacerbate this by optimizing overly for fluency or helpfulness at the expense of faithfulness. For instance, a model might learn that "confident-sounding answers" are more highly rewarded, even if they are incorrect.

3. **Inference process.** At inference time, several additional factors contribute to hallucinations. Ambiguity in prompt formulation can push the model toward unintended interpretations. The randomness of decoding (e.g., sampling strategies with temperature and top-$p$) can generate rare but spurious completions. The so-called *softmax bottleneck* (Yang et al., 2018) can lead the model to overestimate the likelihood of high-probability but incorrect sequences. Finally, reasoning failures in multi-step generation (such as errors in chain-of-thought) can compound uncertainties into hallucinations.

Taken together, these observations highlight that hallucinations and uncertainty are not independent phenomena but two sides of the same coin. Both arise from imperfect data, limited model capacity, and inference variability. While uncertainty reflects the model's recognition of its own limitations, hallucination represents the outward manifestation of those limitations as erroneous outputs. Understanding semantic uncertainty, therefore, is key to diagnosing and mitigating hallucinations in LLMs and LVLMs.

## 3 SEMANTIC-LEVEL UNCERTAINTY

As we have discussed in earlier sections, the uncertainty of LLMs and LVLMs cannot be fully understood at the token level. Token probabilities and token entropies capture local variability in word prediction, but they miss the broader picture of whether the overall *meaning* of the output is stable, accurate, or aligned with the input and the task. This raises an important research question: how can we measure uncertainty in terms of *semantic meaning*?

### 3.1 TWO TYPES OF SEMANTIC UNCERTAINTY

The notion of semantic entropy has also been explored in the context of text-to-image genera-tion (Franchi et al., 2025). Semantic uncertainty is not monolithic. It can be decomposed into at least two main categories:

**Uncertainty due to prompt formulation.** LLMs and LVLMs are highly sensitive to the exact phrasing of a query. For instance, the question *"What is the capital of France?"* may reliably return the answer *"Paris."* But a rephrased prompt such as *"Could you tell me the French capital city?"* or *"What city is the government of France located in?"* may occasionally trigger different responses, formatting issues, or even incorrect answers in low-resource languages (Sclar et al., 2023). This type of uncertainty arises not from the underlying task but from the model's brittleness with respect to prompt wording. This type of uncertainty is tied to stochasticity in the model's response generation and sensitivity to prompt wording. This uncertainty due to unambiguous formatting of the input is linked with semanatic aleatoric uncertainty.

**Uncertainty due to task and model limitations.** The second type of semantic uncertainty is not due to noisy input phrasing but to fundamental limitations of the learned representations. Hence, it comes from the limits of the model's knowledge and reasoning capacity. For example, if a medical image is blurry, the correct diagnosis may be genuinely uncertain, no matter how the question is phrased. Similarly, when the task requires information that lies outside the model's training data (e.g., a newly discovered scientific fact), or when multi-step reasoning introduces compounding errors, the model may exhibit high semantic uncertainty. This uncertainty arises from limitations in the model's knowledge or reasoning, hence, it is linked with semantic epistemic uncertainty, even when the underlying task is unambiguous.

Together, these two types of semantic uncertainty reflect complementary dimensions: *How we ask* (prompt sensitivity) and *what the model can know or reason about* (knowledge and task-related uncertainty). Capturing both dimensions is essential for a full understanding of model reliability.

### 3.2 MODELING SEMANTIC UNCERTAINTY

For a given query $x^{\text{txt}}$ (and image $x^{\text{img}}$ in LVLMs), an LLM produces an output sequence of to-kens $y = [y_1, \ldots, y_T]$, with probabilities $p(y_t \mid y_{<t}, x, \omega)$. Classic token-level entropy captures uncertainty at the word level, but it ignores whether different sequences share the same meaning.

To reason about uncertainty at the level of meaning, we consider distributions over *semantic con-cepts C*. This requires grouping multiple generations into clusters of equivalent meaning and then estimating probabilities over these clusters.

In practice, semantic clustering is challenging and crucial: it transforms open-ended text generation into a prediction over a finite set of possible meanings. We provide details of clustering methods and estimation procedures in Appendix A.1. After clustering, we obtain a set of clusters $\{C_k\}_{k=1}^{K}$, each representing a distinct semantic interpretation of the input. From these clusters, we can derive a distribution over meanings, $p(C_k \mid x, \omega)$.

## 3.3 SEMANTIC ENTROPY

Once the distribution $p(C_k \mid \boldsymbol{x}, \boldsymbol{\omega})$ is estimated, semantic uncertainty can be quantified through *semantic entropy*:

$$H_{\text{SE}}(C \mid \boldsymbol{x}, \boldsymbol{\omega}) = -\sum_{k=1}^{K} p(C_k \mid \boldsymbol{x}) \log p(C_k \mid \boldsymbol{x}). \tag{1}$$

This measure quantifies how dispersed the model's predictions are across distinct meanings. If most probability mass is concentrated in one cluster, semantic entropy is low. If the mass is spread across many clusters, semantic entropy is high.

This measure generalizes classic predictive entropy. In closed-world classification, clusters align with predefined classes $\{O_k\}$, so semantic entropy reduces to $H(Y \mid \boldsymbol{x})$. Extensions include energy-based scoring (Ma et al., 2025). Appendix B further explores the link with mutual information and semantic Uncertainty.

## 3.4 SEMANTIC UNCERTAINTY AND CHAIN OF THOUGHT

Semantic uncertainty is not only a matter of isolated predictions, but also closely linked to the stability of reasoning in CoT. When reasoning unfolds step by step, each additional context element can modify how uncertainty is distributed across the model's outputs. In this subsection we formalize how adding context affects entropy-based and mutual-information-based uncertainty criteria, and then explain how this connects with chain-of-thought reasoning.

### 3.4.1 ENTROPY-BASED UNCERTAINTY CRITERIA WITH MORE CONTEXT

Entropy is a classic measure of uncertainty.

**Lemma 3.1** (Majorization and entropy decrease)**.** Let

$$q = (q_1, \ldots, q_K), \qquad q_k = p(C_k \mid \boldsymbol{x}^{\text{txt}}, \boldsymbol{\omega})$$

and

$$r = (r_1, \ldots, r_K), \qquad r_k = p(C_k \mid \boldsymbol{x}^{\text{txt}}, \boldsymbol{x}_2^{\text{context}}, \boldsymbol{\omega})$$

be two probability vectors on the same finite support, with components arranged in non-increasing order: $q_1 \geq q_2 \geq \cdots \geq q_K$ and $r_1 \geq r_2 \geq \cdots \geq r_K$. If the distribution $r$ *majorizes* the distribution $q$ (denoted $r \succ q$), i.e.,

$$\sum_{i=1}^{m} r_i \geq \sum_{i=1}^{m} q_i \quad \text{for } m = 1, \ldots, K-1, \qquad \text{and} \qquad \sum_{i=1}^{K} r_i = \sum_{i=1}^{K} q_i,$$

then the semantic entropy does not increase:

$$H\big(C \mid \boldsymbol{x}^{\text{txt}}, \boldsymbol{x}_2^{\text{context}}, \boldsymbol{\omega}\big) = H(r) \leq H(q) = H\big(C \mid \boldsymbol{x}^{\text{txt}}, \boldsymbol{\omega}\big).$$

*Proof.* We rely on a standard fact from majorization theory: if $f$ is convex on an interval containing the probability simplex, then the function $p \mapsto \sum_i f(p_i)$ is *Schur-convex*; equivalently, its negative is Schur-concave. Hence the map

$$p \longmapsto -\sum_{i=1}^{K} \phi(p_i)$$

is Schur-concave. Since the Shannon entropy can be written as

$$H(p) = -\sum_{i=1}^{K} p_i \log p_i = -\sum_{i=1}^{K} \phi(p_i).$$

Therefore $H(\cdot)$ is Schur-concave. By the defining property of Schur-concavity,

$$r \succ q \implies H(r) \leq H(q).$$

This is exactly the desired inequality. $\qquad\square$

**Remark.** Intuitively, $r \succ q$ means the distribution $r$ is more concentrated (less spread) than the distribution $q$. Hence, that means that the context must be a context that reduces spread of $q$.

In other words: adding relevant context to the prompt can only reduce the model's semantic uncertainty. For example, if the prompt is *"What is happening in this picture?"* and the additional context specifies *"Focus on the left side of the image"*, then the entropy of possible outputs decreases, since the model has less ambiguity about which region to describe.

### 3.4.2 CHAIN-OF-THOUGHT AS AN UNCERTAINTY REDUCER

Now consider chain-of-thought reasoning, where the model generates intermediate steps $Z_1, Z_2, \ldots, Z_T$. Each step can be viewed as an additional piece of context, analogous to $\boldsymbol{x}_t^{\text{context}}$.

**Property 1** (Chain-of-Thought Reduces Entropy)**.** If we iteratively enrich a prompt with contexts $(Z_1, Z_2, \ldots, Z_T)$, where the contexts follow the same majorization property as in Lemma 3.1, then repeated application of Lemma 3.1 implies that the entropy of the model's semantic output satisfies

$$H(C \mid \boldsymbol{x}^{\text{img}}, \boldsymbol{x}_1^{\text{txt}}, \boldsymbol{\omega}) \ \geq \ H(C \mid \boldsymbol{x}^{\text{img}}, \boldsymbol{x}_1^{\text{txt}}, Z_1, Z_2, \ldots, Z_T, \boldsymbol{\omega}).$$

In other words, chain-of-thought reasoning can be understood as a structured way of *progressively reducing uncertainty* by conditioning on intermediate reasoning steps. This provides a formal justification for why CoT prompting often improves reliability in LLMs and LVLMs: by exposing intermediate reasoning, we effectively add context that reduces entropy and sharpens the model's semantic predictions.

## 4 EMPIRICAL VALIDATION AND LIMITATIONS

We conduct a small experiment using **Llama-3.1-8B-Instruct** on the **TriviaQA** dataset (Joshi et al., 2017). TriviaQA is a large-scale reading comprehension dataset containing over 650k question-answer pairs, collected from trivia enthusiasts and paired with evidence documents from Wikipedia and the web. It is widely used to evaluate open-domain question answering and reasoning.

In our setup, the model is queried **20 times per question**, and we compare two semantic clustering strategies: one based on **GPT-4o** and one based on **DeBERTa**. We then evaluate the quality of uncertainty estimates using the **Expected Calibration Error (ECE)** and the **AUROC**. For both metrics, we first check whether the model's output is correct. We then normalise the uncertainty score so that it lies between 0 and 1 before calculating the metrics. For token-level uncertainty, we use the **perplexity** (Brown et al., 1992), which measures how well a language model predicts the next word in a sequence. In simple terms, a low perplexity means the model is confident and accurate in its predictions, while a high perplexity means the model is uncertain or surprised by the actual next word.

Figure 2 shows that **perplexity-based uncertainty is poorly calibrated**, a result confirmed in Table 1. We observe that **perplexity yields poor calibration** (ECE of 31.5%) and weak discrimination (AUROC of 54.1%). In contrast, **semantic entropy provides a substantial improvement**, with DeBERTa-based clustering reducing ECE to 22.7% and increasing AUROC to 79.2%. The **best results are obtained with GPT-4o-based clustering**, which achieves both the lowest ECE (13.5%) and the highest AUROC (80.9%). Hence, GPT-4o clustering achieves better calibration than DeBERTa, but this comes at a cost: the method introduces randomness, with about **2% standard deviation** on both ECE and AUROC. This variability arises from the stochastic nature of the clustering strategy. Although this difference may seem small in terms of accuracy, it can have a strong impact in uncertainty-sensitive applications.

Finally, we note that semantic uncertainty estimation is computationally expensive, since it requires running a large model such as GPT-4o to compute the clustering.

**Limitations of Semantic Uncertainty.** The limitations of semantic uncertainty can be grouped into three main points. First, estimating uncertainty requires generating multiple answers, but it is unclear how many are needed for a reliable estimate without incurring high computational cost. Second, clustering introduces its own uncertainty: it can be stochastic, depends on another LLM, and lacks a universal ground truth. Human-annotated clusters may be useful to better assess quality.

Table 1: **Comparison of uncertainty estimation methods on TriviaQA.** Lower ECE indicates better calibration, while higher AUROC indicates better discrimination between correct and incorrect answers.

|  | ECE (%) | AUROC (%) |
|---|---|---|
| Perplexity | 31.50 | 54.07 |
| Semantic Entropy (DeBERTa) | 22.66 | 79.16 |
| Semantic Entropy (GPT-4o) | 13.53 | 80.94 |

Third, current methods mainly apply to short answers, and it is unknown how well they generalize to other tasks. Finally, recent work on *verbalized uncertainty* (Ji et al., 2025) stresses the need to communicate model uncertainty effectively to humans, who are often in the decision loop.

## 5 CONCLUSION.

Foundation models such as LLMs and LVLMs have transformed the field of AI, yet their remarkable capabilities come with fundamental challenges in reasoning under uncertainty. Token-level confidence measures, while effective in closed-world settings, are inadequate in open-world or multimodal scenarios: they fail to capture *semantic-level uncertainty*, which is essential for detecting and interpreting hallucinations. In settings where outputs are unbounded and often disconnected from ground truth, assessing the reliability of *meaning* rather than surface tokens becomes indispensable.

In this paper we we introduce a taxonomy for uncertainty quantification in LLMs and LVLMs and propose formalism for analyzing how uncertainty. Although semantic uncertainty comes with weaknesses—such as entropy instability under sampling, dependence on representation models, and variability due to prompt formulation—it provides a direct way to assess the reliability of LLM and LVLM outputs, and it is a key tool for detecting hallucinations that cannot be identified through token-level metrics alone.

We view this work as a step toward a broader research agenda: building principled methods for semantic uncertainty quantification, grounding them in empirical evaluation, and designing ways to communicate uncertainty effectively to human users. Addressing these challenges is essential if foundation models are to be deployed safely and reliably in open-world applications. Reasoning about uncertainty in foundation models is difficult, particularly in open-world settings where the space of possible outputs is unbounded. Yet the stakes are high: safe deployment of LLMs and LVLMs in critical applications requires not only stronger methods for uncertainty quantification, but also ways of communicating this uncertainty effectively to human users.

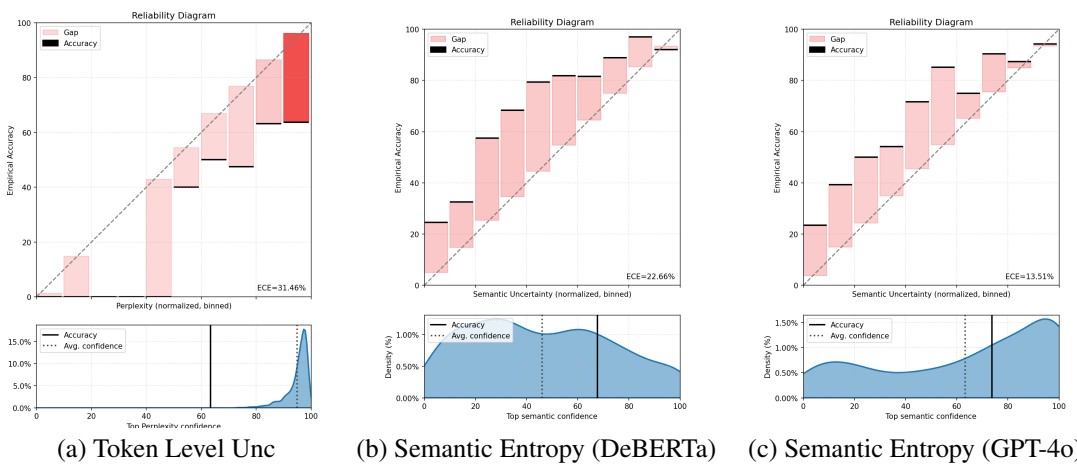

(a) Token Level Unc     (b) Semantic Entropy (DeBERTa)     (c) Semantic Entropy (GPT-4o)

Figure 2: Calibration plots for different uncertainty criteria: **(a)** Perplexity, **(b)** Semantic Entropy with DeBERTa, and **(c)** Semantic Entropy with GPT-4o.

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

# A   SEMANTIC UNCERTAINTY

## A.1   MODELING SEMANTIC UNCERTAINTY

Let $\boldsymbol{x}^{\text{txt}}$ denote a natural language query provided to an LLM. In the LVLM case, the input also contains an image $\boldsymbol{x}^{\text{img}}$. For simplicity of notation, we focus on LLMs, but all definitions extend directly to LVLMs.

Given an input, the model produces an output sequence of tokens:

$$\boldsymbol{y} = [y_1, y_2, \ldots, y_T], \tag{2}$$

where $T$ is the output length.

The model assigns probabilities to each token conditioned on the previous tokens and the input query:

$$p(y_t \mid y_{<t}, \boldsymbol{x}^{\text{txt}}, \boldsymbol{\omega}). \tag{3}$$

The classic token-level entropy at position $t$ is:

$$H_t = -\sum_{y \in \mathcal{V}} p(y \mid y_{<t}, \boldsymbol{x}^{\text{txt}}, \boldsymbol{\omega}) \log p(y \mid y_{<t}, \boldsymbol{x}^{\text{txt}}, \boldsymbol{\omega}), \tag{4}$$

where $\mathcal{V}$ is the vocabulary.

This measure, however, does not tell us whether two different token sequences correspond to the same semantic meaning. To capture uncertainty at the level of meaning, we instead seek a distribution over *concepts*, written as:

$$p(C \mid \boldsymbol{x}, \boldsymbol{\omega}),$$

where $C$ is a random variable representing the semantic content of the model's response.

## A.2   STEPS TO APPROXIMATE SEMANTIC UNCERTAINTY

Direct access to $p(C \mid \boldsymbol{x}, \boldsymbol{\omega})$ is typically not available. Therefore, researchers approximate it through a three-step procedure (Farquhar et al., 2024; Qiu & Miikkulainen, 2024; Nikitin et al., 2024; **?**; Grewal et al., 2024; Kuhn et al., 2023):

**Step 1: Generate multiple responses.**   We produce $n$ different generations for the same input, for example, by sampling with temperature or nucleus sampling:

$$\{\boldsymbol{y}^{(1)}, \boldsymbol{y}^{(2)}, \ldots, \boldsymbol{y}^{(n)}\}, \quad \boldsymbol{y}^{(i)} = [y_1^{(i)}, y_2^{(i)}, \ldots, y_{T_i}^{(i)}]. \tag{5}$$

**Step 2: Cluster responses by meaning.**   The generated outputs are grouped into semantic clusters, such that responses with equivalent meanings belong to the same cluster. Several strategies exist:

- One widely used strategy for clustering generations by meaning relies on Natural Language Inference (NLI) models, in particular DeBERTa (He et al., 2020; Farquhar et al., 2024; Kuhn et al., 2023). The idea is to use the model's ability to judge whether one sentence entails another. Formally, given two generated outputs $\boldsymbol{y}^{(i)}$ and $\boldsymbol{y}^{(j)}$, they use DeBERTa to predict whether the statement $\boldsymbol{y}^{(i)}$ *entails* $\boldsymbol{y}^{(j)}$, and vice versa. If $\boldsymbol{y}^{(i)}$ entails $\boldsymbol{y}^{(j)}$ *and* $\boldsymbol{y}^{(j)}$ entails $\boldsymbol{y}^{(i)}$, then the two outputs are judged to have the same semantic meaning. This condition, called *bi-directional entailment*, ensures that both sentences are not only compatible but effectively equivalent in meaning.

  Clustering is then built incrementally:

  - For each new generated output $\boldsymbol{y}^{(i)}$, they check whether bi-directional entailment holds with *all members* of an existing cluster $C_k$.
  - If so, $\boldsymbol{y}^{(i)}$ is added to that cluster.
  - If no cluster satisfies this condition, $\boldsymbol{y}^{(i)}$ starts a *new* cluster, representing a distinct semantic meaning.

After all generations are processed, they obtain a set of clusters $\{C_k\}_{k=1}^K$, where each cluster corresponds to one unique semantic interpretation of the input.

The limitation of this approach is that this procedure enforces *hard clustering*: each output is either fully inside or outside a cluster.

- Recent work (Chen et al., 2025) extends this idea by using DeBERTa predictions to construct a semantic similarity graph between outputs. In this graph, edges are weighted by entailment scores, and clustering is performed using soft methods that allow partial membership. This produces clusters that better reflect nuanced overlaps in meaning, where an output may be semantically close to multiple groups rather than strictly belonging to one.

- Use kernel methods to avoid explicit clustering (Nikitin et al., 2024) and also avoid estimate $p(C \mid \boldsymbol{x}, \boldsymbol{\omega})$.

- Apply lexical similarity combined with improved clustering models (Chen et al., 2024; Nguyen et al., 2025).

- Utilize Inter- vs. intra-cluster separation to improve the clustering (Joo & Cho, 2025).

- Ask an LLM itself to cluster outputs by meaning (Farquhar et al., 2024; Ji et al., 2025).

After this step, we obtain $K$ semantic clusters $\{C_k\}_{k=1}^K$, each representing one possible interpretation. Interestingly, clustering effectively transforms an open-ended generation problem into a closed-world prediction problem over a finite set of meanings.

**Step 3: Estimate cluster probabilities.** The probability of an individual response is:

$$p(\boldsymbol{y}^{(i)} \mid \boldsymbol{x}, \boldsymbol{\omega}) = \prod_{t=1}^{T_i} p(y_t^{(i)} \mid y_{<t}^{(i)}, \boldsymbol{x}, \boldsymbol{\omega}). \tag{6}$$

This can be normalized across samples:

$$\bar{p}(\boldsymbol{y}^{(i)} \mid \boldsymbol{x}, \boldsymbol{\omega}) = \frac{p(\boldsymbol{y}^{(i)} \mid \boldsymbol{x}, \boldsymbol{\omega})}{\sum_{j=1}^n p(\boldsymbol{y}^{(j)} \mid \boldsymbol{x}, \boldsymbol{\omega})}. \tag{7}$$

The probability of a cluster $C_k$ is then:

$$p(C_k \mid \boldsymbol{x}, \boldsymbol{\omega}) = \sum_{\boldsymbol{y}^{(i)} \in C_k} \bar{p}(\boldsymbol{y}^{(i)} \mid \boldsymbol{x}, \boldsymbol{\omega}). \tag{8}$$

If probabilities are not accessible, frequency-based estimates can be used:

$$p(C_k \mid \boldsymbol{x}, \boldsymbol{\omega}) = \frac{1}{n} \sum_{i=1}^n \mathbf{1}[\boldsymbol{y}^{(i)} \in C_k]. \tag{9}$$

Qiu & Miikkulainen (2024) propose using kernel density estimation to have a better estimate of $p(C_k \mid \boldsymbol{x}, \boldsymbol{\omega})$.

**Why clustering matters.** Accurate clustering is essential: too coarse, and distinct meanings collapse; too fine, and equivalent phrasings appear different. This balance is particularly hard in practice, yet it is key to obtaining meaningful estimates of semantic uncertainty.

## B MUTUAL INFORMATION OVER PROMPTS

Another principled approach to quantifying semantic uncertainty is through *mutual information* (MI). Recall that the mutual information between two random variables $X$ and $Y$ is defined as

$$I(X;Y) = H[X] - \mathbb{E}_{P(y)}\big[H[X \mid Y = y]\big]$$
$$= H[Y] - \mathbb{E}_{P(x)}\big[H[Y \mid X = x]\big],$$

where $H[\cdot]$ denotes Shannon entropy. Intuitively, $I(X;Y)$ measures how much knowing one variable reduces uncertainty about the other.

In the setting of Bayesian neural networks (BNNs), MI has been widely used as an epistemic uncertainty measure. Specifically, the mutual information between the label $y$ of a new input $x$ and the model parameters $\omega$, given training data $\mathcal{D}$, is

$$I(\omega; y \mid \mathcal{D}, x) = H[p(y \mid x, \mathcal{D})] - \mathbb{E}_{p(\omega \mid \mathcal{D})}\Big[H[p(y \mid x, \omega)]\Big]. \tag{10}$$

This expression captures the idea that if the model is uncertain about $x$, then observing its true label $y$ would substantially reduce uncertainty about the parameters $\omega$. Hence, mutual information quantifies the potential information gain.

**From Bayesian Models to LLMs and LVLMs.** For large language models (LLMs) and large vision-language models (LVLMs), however, we cannot directly access the Bayesian posterior $p(\omega \mid \mathcal{D})$. Instead, a key source of variability comes from the distribution of prompts provided to the model. Different textual contexts may highlight different aspects of the same input, leading to semantically different outputs. To capture this phenomenon, we define an analogous measure of semantic uncertainty by marginalizing over prompts:

$$I(C; \boldsymbol{x}^{\text{txt}} \mid \boldsymbol{x}^{\text{img}}, \omega) = H\big[p(C \mid \boldsymbol{x}^{\text{img}}, \omega)\big] - \mathbb{E}_{p(\boldsymbol{x}^{\text{txt}})}\Big[H\big(C \mid \boldsymbol{x}^{\text{img}}, \boldsymbol{x}^{\text{txt}}, \omega\big)\Big]. \tag{11}$$

Here, $C$ denotes the semantic content of the model's output, and the marginal predictive distribution is

$$p(C \mid \boldsymbol{x}^{\text{img}}, \omega) = \mathbb{E}_{p(\boldsymbol{x}^{\text{txt}})}\big[p(C \mid \boldsymbol{x}^{\text{img}}, \boldsymbol{x}^{\text{txt}}, \omega)\big].$$

**Decomposition of Predictive Uncertainty.** Equation equation 11 yields a natural decomposition of the total predictive uncertainty:

$$H(C \mid \boldsymbol{x}^{\text{img}}, \omega) = \underbrace{\mathbb{E}_{p(\boldsymbol{x}^{\text{txt}})}\Big[H(C \mid \boldsymbol{x}^{\text{img}}, \boldsymbol{x}^{\text{txt}}, \omega)\Big]}_{\text{Intrinsic uncertainty given a fixed prompt}} + \underbrace{I(C; \boldsymbol{x}^{\text{txt}} \mid \boldsymbol{x}^{\text{img}}, \omega)}_{\text{Uncertainty due to prompt variability}}. \tag{12}$$

This decomposition separates two conceptually distinct contributions:

- The first term measures the expected entropy of predictions given a fixed prompt. It reflects uncertainty intrinsic to the task (e.g., label ambiguity, inherent noise) once the textual context is fixed.
- The second term is the mutual information, which quantifies the additional uncertainty induced by variability across prompts. This captures how sensitive the model's semantic predictions are to prompt phrasing, thereby isolating uncertainty arising from *contextual dependence*.

