# OpenReview forum: "From Tokens to Meaning: LLMs and LVLMs Require Semantic-Level Uncertainty"
_ICLR.cc/2026/Conference — Submitted to ICLR 2026_

### Official Review · Reviewer_n3dm · 2025-10-21

**Soundness:** 2
**Presentation:** 3
**Contribution:** 1
**Rating:** 2
**Confidence:** 5

**Summary:**

This is a position paper, that argues that the field of LLM uncertainty should be moving away from token uncertainty to semantic uncertainty, and that provides classification of different sources of uncertainty.

I do not feel this is a timely position paper. The first semantic entropy paper was put on arxiv in early 2023, and the uncertainty quantification for LLM community have widely adopted semantic-based approaches as a gold standard (although in practice, other methods are often used for computational reasons). The definitions of uncertainty in Sections 2.2 and 2.3 are all based on prior papers. The CoT-based section seems novel, but not well fleshed out.

**Strengths:**

* The premise is reasonable — IMO semantic clustering is the right way to think about uncertainty in natural language.

**Weaknesses:**

* Much of this is already accepted by the community
* The theoretical contribution is unclear (see below)
* IMO, the single experiment seems to oversell the benefits of semantic entropy (when compared to other, more thorough, evaluations in the literature)

**Questions:**

* There seems to be a big conceptual leap from Lemma 3.1, to the remark under Lemma 3.1. How can we guarantee that relevant information causes majorization? In the example given, if the whole image was a picture of the Eiffel Tower, but the left hand side of the picture contained birds, people, buildings, trees... I would expect greater semantic entropy when asked to focus on the left hand side. Can you elaborate on how and when the remark follows from Lemma 3.1? The statement at the beginning of the remark---“Intuitively, r ≻q means the distribution r is more concentrated (less spread) than the distribution q. Hence, that means that the context must be a context that reduces spread of q."---feels almost like a tautology to me: If I have a context that reduces the entropy of q, then the entropy will reduce.
* Property 2 feels similarly tautological: If reasoning steps in chain of thought reduce uncertainty about the output, then uncertainty will reduce. It certainly isn’t a golden rule that uncertainty monotonically reduces during reasoning.
* It’s unclear to me how you calculate ECE for either Semantic Entropy, or perplexity. Calibration (at least, of the form measured using ECE) is a property of a number between zero and one; neither SE nor perplexity fit this description. Can you elaborate how you calculate SE?
* The AUROC results give a much larger difference between SE and perplexity/other logit-based methods than I have seen in other papers (eg, Aichberger et al, 2023; Fadeeva et al, 2023; Santilli et al; 2025. How is correctness calculated? Santilli et al (2025) show that using length-biased correctness metrics can lead to underestimating the AUROC of perplexity.
    * It would be good to see a more comprehensive evaluation here, with more datasets/more logit-based methods. I agree that Semantic Entropy is typically better than logit-based methods, but I feel the results here oversell the difference.


Aichberger et al, Rethinking Uncertainty Estimationin Natural Language Generation. Arxiv 2024.
Fadeeva et al, LM-polygraph: Uncertainty estimation for language models.. EMNLP 2023.
Santilli et al, Revisiting Uncertainty Quantification Evaluation in Language Models: Spurious Interactions with Response Length Bias Results. ACL, 2025

---

### Official Review · Reviewer_8jsd · 2025-10-28

**Soundness:** 2
**Presentation:** 3
**Contribution:** 1
**Rating:** 2
**Confidence:** 2

**Summary:**

The paper proposes to look beyond token-level uncertainty and instead define uncertainty at a semantic sequence level. This is due to issues with token-level uncertainty such as ignoring *verbalized uncertainty* which might still have a large per-token likelihood of being generated. The authors define multiple sources of general uncertainty and two types of semantic uncertainty. Finally, the authors conduct a small experiment using a 8B model on TriviaQA using semantic entropy.

**Strengths:**

The paper correctly argues for going beyond token-level uncertainty and provides an overview of relevant works in this area.

**Weaknesses:**

W1: I am really unclear about the novelty of this work.
- What is the difference between the work on Semantic Entropy [Farquhar et al., 2024] and the "new" semantic entropy / semantic uncertainty defined in Section 3 and used for the experiments in Section 4?
  - The paper does indeed cite Farquhar et al., 2024 but then continues to define "semantic entropy" and "semantic uncertainty" in very similar ways without any references to their original work.
- If the paper's contribution is not new methodology but a new framework of looking at uncertainty, how is the papers new approach of advocating for "sequence-level uncertainty" different from the points already made by Farquhar et al., 2024 (e.g. quote from page 2: "Naive entropy-based uncertainty measures variation in the exact answers, treating ‘Paris’, ‘It’s Paris’ and ‘France’s capital Paris’ as different. But this is unsuitable for language tasks for which sometimes different answers mean the same things. Our semantic entropy clusters answers which share meanings before computing the entropy.")?

**Questions:**

Q1: l.227ff ("Factuality hallucinations.") -- Is the example given (“Who is the Chancellor of Germany in 2025?”, an LLM might confidently reply “Olaf Scholz”, even though the mandate of Olaf Scholz has ended.") really the best here? To me this would be a different type of "hallucination" (I would actually argue for a different term in this case), since the given reply could be entirely consistent with the pretraining corpus. This is markedly different from hallucinated replies, which have no grounding at all in the training data but are the results of a lack of "abstaining" in case of insufficient knowledge to answer a query.

---

### Official Review · Reviewer_P8cW · 2025-10-31

**Soundness:** 2
**Presentation:** 4
**Contribution:** 2
**Rating:** 2
**Confidence:** 4

**Summary:**

This position paper argues that token-level uncertainty is fundamentally inadequate for assessing reliability in large language and vision-language models (LLMs and LVLMs). The authors contend that meaningful uncertainty quantification must shift from surface-level token probabilities toward semantic-level uncertainty—uncertainty in meaning rather than in form.
They formalize semantic entropy as the entropy of the model’s distribution over latent meanings, weighted by model likelihoods across semantically clustered generations, extending the count-based formulation of Kuhn et al. (2023).
They present a conceptual taxonomy linking hallucination, uncertainty, and reasoning, derive theoretical properties (e.g., entropy majorization under chain-of-thought conditioning), and provide a small empirical demonstration on TriviaQA comparing token-level perplexity with semantic entropy computed via DeBERTa and GPT-4o clustering.

**Strengths:**

1. Highly motivated: highlights a genuine limitation of token-level entropy in capturing semantic ambiguity and reasoning uncertainty.
2. Conceptually clear: presents a clean taxonomy of uncertainty sources across LLMs and LVLMs.
3. Readable and well-organized: writing and presentation are clear.
4. A small experiment supports the main claim that semantic-level entropy better correlates with factual correctness.

**Weaknesses:**

1. Incremental formulation: the proposed semantic-entropy definition is a modest extension of Kuhn et al. (2023)—essentially reweighting clusters by model likelihoods rather than counts. While conceptually tidy, it does not constitute a fundamentally new measurement framework.
2. Limited empirical scope: evaluation is minimal (single dataset, single model) and relies on external entailment models for clustering.
3. Multimodal aspect underdeveloped: LVLM discussion is theoretical; no experiments substantiate the multimodal claims.
4. Formal results are illustrative: the majorization and reasoning analyses remain qualitative without quantitative validation.

**Questions:**

1. How does the proposed weighted semantic entropy behave when token-probability variance is high but all samples share the same meaning (single-cluster case)?
2. Are there LVLM settings where token-level entropy still correlates with semantic uncertainty?
3. Could semantic entropy be estimated directly from hidden representations instead of clustering via external entailment models?
4. How sensitive are the results to the specific entailment model used for clustering (DeBERTa vs. GPT-4o)?

---

### Official Review · Reviewer_aCTG · 2025-11-05

**Soundness:** 1
**Presentation:** 1
**Contribution:** 1
**Rating:** 2
**Confidence:** 4

**Summary:**

The authors argue that the uncertainty of large language models (LLMs) and large vision-language models (LVLMs) should be expressed on a semantic level rather than a token level. They also suggest semantic uncertainty should be the basis of assessing these models’ reliability.

**Strengths:**

Originality: the empirical results presented in Section 4 are new.

Quality: the authors have made a good effort to aggregate ideas around LLM/LVLM uncertainty.

Clarity: the paper is overall easy to follow.

Significance: LLM/LVLM uncertainty is a topic that warrants research attention.

**Weaknesses:**

The authors frame this as a position paper, but even with that framing the contribution is limited. The position itself is essentially a rehash of existing work (Farquhar et al, 2024; Kossen et al, 2024; Kuhn et al, 2023), and insofar as the authors go further in promoting semantic uncertainty as a tool—they write “semantic uncertainty should become the primary lens through which we
assess the reliability of foundation models” in their abstract—they actually make a dangerous association between subjective uncertainty and reliability (Bickford Smith et al, 2025). The content supporting the authors’ three stated contributions is also rather weak:

- Contribution 1 is “We propose a taxonomy for uncertainty quantification in LLMs and LVLMs and explicitly link it to hallucination”, which presumably refers to Section 2. While there can be value in aggregating various observations from past work, as the authors do, I think it’s a stretch to call this a taxonomy rather than just a literature review.
- Contribution 2 is “We introduce a formalism to analyze how uncertainty evolves with the context provided to the model”, which refers to Section 3.4. I think this section is misleading. Without basing their “majorisation” assumption in reality, I don’t think they have a grounds to say “adding relevant context to the prompt can only reduce the model’s semantic uncertainty”. Indeed, it appears their takeaway is in direction contradiction with their earlier claim that “Multi-step reasoning tasks amplify uncertainty”. Arguably most importantly, the authors end the section with an incorrect association (Bickford Smith et al, 2025) between uncertainty reduction and reliability.
- Contribution 3 is “We discuss the limitations of semantic uncertainty and outline directions for future work”, which refers to the final paragraph in Section 4. This is potentially where the authors had the greatest opportunity to contribute, but really this amounts to four points, three of which comprise a single sentence.

---

Bickford Smith et al (2025). Rethinking aleatoric and epistemic uncertainty. ICML.

Farquhar et al (2024). Detecting hallucinations in large language models using semantic entropy. Nature.

Kossen et al (2024). Semantic entropy probes: robust and cheap hallucination detection in LLMs. Workshop on Foundation Models in the Wild, ICML.

Kuhn et al (2023). Semantic uncertainty: linguistic invariances for uncertainty estimation in natural language generation. ICLR.

**Questions:**

Can you relate your theory in Section 3.4 to models’ empirical behaviour?

Can you expand on your discussion of semantic uncertainty’s current limitations, sketching out a reasoned position on concrete directions that the field should explore in future work?

---

### Official Review · Reviewer_2rWA · 2025-11-06

**Soundness:** 1
**Presentation:** 1
**Contribution:** 1
**Rating:** 0
**Confidence:** 4

**Summary:**

Authors express their opinion that Semantic-level uncertainties are important.

**Strengths:**

S1. I agree with alnost everything authors say up to the end of Sec 3.

**Weaknesses:**

W1. Lack of technical depth.
In my opinion, this paper lacks technical depth expected of a paper published at venue like ICLR.
I appreciate position papers might be important, but in my opinion, this paper does not present ideas controversial to anyone in the audience (the concrete subfield of UQ for LLMs/VLMs), neither does it present them with the level of rigour or depth (e.g. proposing an valueable new taxonomy or viewpoint) that would be of value to the reader that would warrant acceptance in the proceedings.
In my opinion, the importance of using semantic uncertainty has already been acknowledged by the field as evidenced by the mounting number of works studying the subject, starting with the "seminal" work of Kuhn et al.


W2. Vacuous technical rigour in Sec 3.4.
The result in this section serves no purpose in the context of the entire paper - they are not utilized in any way, nor verified whether they hold empirically.

W3. Empirical evaluation of Sec 4 doesn't contribute anything.

W3a. The results of Sec 4 are completely redundant given the empirical evaluations of previous works, e.g. https://github.com/AlexanderVNikitin/kernel-language-entropy

W3c. Normalizing a semantic entropy score to [0, 1] range and evaluating calibration-error makes no sense from the point of view of what those quantities signify.

**Questions:**

N/A

---

### Meta-Review · Area_Chair_2Afb · 2025-12-05

**Summary:**

This paper argues that token-level uncertainty is insufficient for evaluating the reliability of large language and vision-language models. Instead, uncertainty should be defined at the semantic level, capturing uncertainty in meaning rather than surface token probabilities. The paper formalizes semantic entropy as the entropy of a model’s distribution over latent meanings, obtained by clustering multiple generated responses based on semantic similarity and weighting them by model likelihoods. They discuss how semantic uncertainty relates to hallucination and reasoning, outline theoretical properties, and provide a small experiment on TriviaQA demonstrating that semantic entropy better reflects model confidence than token-level perplexity.

As most reviewers pointed out, this work is mainly a position paper and the technical contributions remain limited. In addition, several concerns raised by the reviewers were not sufficiently addressed during the rebuttal. Therefore, I do not believe the paper can be accepted in its current form. I encourage the authors to strengthen the technical content and resubmit to a future venue.

**Reviewer Scores:**

The authors did not respond to their reviews. So, I think reviewers will maintain their original score.

---

### Decision · Program_Chairs · 2026-01-26

Reject